# Exposure image correction of electrical equipment nameplate based on the LMPEC algorithm

Hao Wu[1,2☯]*, Yanxi Liu[1,2☯], Zhongyang Jin[1,2‡], Yuan Zhou[1,2‡]

1 Automation and Information Engineering, Sichuan University of Science & Engineering, Yibin, Sichuan, China, 2 Artificial Intelligence Key Laboratory of Sichuan Province, Yibin, Sichuan, China

☯ These authors contributed equally to this work.
‡ ZJ and YZ also contributed equally to this work.
* 11305076@qq.com

**Data Availability Statement:** All relevant data are within the manuscript and its Supporting information files.

**Funding:** This study received partial funding from the Sichuan Provincial Department of Science and

## Abstract

An optimization algorithm based on the LMPEC algorithm is proposed to rectify the nameplate image to address the problem that overexposure and underexposure of the nameplate image of electrical equipment will make subsequent nameplate recognition difficult. In the network structure, the PS-UNet++ network is based on the sub-pixel convolution upsampling module, and the UNet++ network is constructed as the feature extraction sub-network of the optimization algorithm to extract more detailed information from the model. Smooth L1 loss is substituted for L1 loss in the loss function to prevent model oscillation. In addition, to increase the robustness of the model, an improved method built on the multi-scale training method is applied. The experimental results indicate that, among all comparison algorithms, the optimized algorithm performs the best on the data set of electrical equipment nameplate exposure the experimenter generated. Compared to the original LMPEC algorithm, the SSIM, PSNR, and PI image evaluation indices are enhanced by 5.6%, 5.1%, and 7.96%, respectively.

## Introduction

China's power infrastructure is extensive in scale and has many equipment. Generally, it is necessary to record the equipment information on the nameplate to manage the power system equipment. Currently, the information registration of electrical equipment nameplates is completed predominantly manually, which is not only inefficient and prone to error but also a significant human resource drain. Therefore, establishing a registration method for intelligent electrical equipment nameplates is crucial for developing a smart power grid [1, 2]. In the power system, some electrical equipment needs to be installed outdoors. Their nameplates are usually metal, leading to overexposure or underexposure issues in some captured nameplate images. Due to the large amount of text information and relatively small targets on the nameplate of electrical equipment, detecting the text information on the nameplate is challenging. However, the problem of overexposure or underexposure on the nameplate further increases the difficulty of subsequent detection and recognition of electrical equipment nameplate

Technology projects 2021YFG0313 and 2022YFS0518.

**Competing interests:** The authors have declared that no competing interests exist.

information [3]. Electrical equipment nameplate images with good exposure have higher detection accuracy and fewer missed detections in text detection tasks than those with insufficient or excessive exposure. At the same time, it also has better recognition accuracy in text recognition tasks. Solving the exposure problem of electrical equipment nameplate images is beneficial for detecting and recognizing electrical equipment nameplate information. This paper examines the image correction of electrical equipment nameplates under overexposure and underexposure conditions.

Histogram equalization (HE) is frequently employed as the primary solution to the exposure problem. This method improves image contrast by expanding the intensity distribution range of pixels to rectify exposure [4]. However, the histogram equalization method will increase background noise and reduce the difference of local information, resulting in an image with excessive or inadequate correction. Then, in response to the deficiencies of histogram equalization in coping with image exposure, several researchers proposed the local adaptive histogram equalization (CLAHE) algorithm [5]. Although CLAHE is more effective than HE at correcting overexposed images, it is still inadequate at correcting images with excessively brilliant or dark exposure.

Land proposed the Retinex theory in the 1960s, which held that images comprise the illumination component I and the reflection component R [6]. Based on this, Chen et al. combined deep learning with Retinex theory and proposed the Retinexnet image enhancement algorithm, which uses a convolutional neural network to process the image to obtain more reasonable illumination and reflection components [7], and the correction effect of this method on the exposed image is also superior to that of the HE. Then, Zhang et al. proposed a high-quality exposure correction (HQEC) algorithm based on the Retinex theory, which obtained a better correction effect by employing perceptual bidirectional similarity (PBS) to constrain the exposure image in terms of exposure, color, and texture [8]. Even though the image enhancement algorithm based on Retinex theory has been developed for many years, it cannot automatically adjust the parameters based on the image quality, resulting in a complex algorithm and color loss in the final corrected image.

In recent years, some researchers have proposed and successfully implemented novel algorithms for image exposure. Guo et al., for instance, proposed the zero-reference deep curve estimation (Zero-DCE) algorithm in conjunction with the notion of self-monitoring [9]. The benefit of this algorithm is that, during training, the model does not require coupled image data sets as a reference. It transforms the problem of weak light enhancement into a curve transformation problem and utilizes the network simulation curve to guide iteration and iterative enhancement. Ma et al. proposed a rapid and adaptable network for underexposure correction [10]. The network model proposed a self-calibration lighting learning module with weight sharing, which enhanced exposure stability, decreased model calculation volume, and yielded positive results. The learning multi-scale photo exposure correction algorithm (hereafter referred to as LMPEC) was proposed by Afifi et al. [11]. The algorithm simultaneously corrects overexposed and underexposed images using a single-depth learning model. It divides the exposure correction problem into two parts, color enhancement and detail enhancement, and processes each separately to produce a decent exposure-corrected image. It is a superior algorithm for correcting overexposed and underexposed images simultaneously.

## Contributions

This paper optimizes the overexposure and underexposure problems of the nameplate image of electrical equipment using the LMPEC algorithm proposed by Afifi et al. In terms of model structure, firstly, a PS-UNet++ network based on sub-pixel convolution and UNet++ network

is constructed. Then, the PS-UNet++ network is used to replace the three-layer and four-layer UNet-like subnetworks used for encoding and decoding in the LMPEC algorithm to improve the feature extraction ability of the model. At the same time, optimizations have been made to the loss function and training method to improve the model's correction effect on uneven exposure nameplates further. In addition, this paper also established an uneven exposure data-set for electrical equipment nameplates, which includes 5000 images of electrical equipment nameplates under different exposure parameter settings. Finally, this paper conducts ablation and comparative experiments on the proposed method to demonstrate the effectiveness of the optimization algorithm.

## LMPEC algorithm

### Model structure

LMPEC algorithm proposes a coarse-to-fine depth learning algorithm that addresses the over-exposure and underexposure problems in sRGB images. The algorithm separates the image exposure correction problem into two components: color enhancement and detail enhance-ment. The model of the deep neural network is utilized for end-to-end training. After correct-ing the image's global color information, the image's detail information is enhanced. Fig 1 depicts the LMPEC algorithm's model structure.

The model structure diagram of the LMPEC algorithm reveals that the to-be-corrected nameplate image is first decomposed into n images using the Laplace pyramid. These images mainly store the information in upper and lower sampling, ensuring that too much detailed information will not be lost during the information exchange between high-resolution and low-resolution images [12]. Afterward, the image output from the *n-th* layer of the Laplacian pyramid will be input into the first sub-network of the LMPEC algorithm, a 4-layer UNet-like network primarily used for feature extraction. The image processed by the first sub-network will be upsampled, and the result will be added to the Laplacian pyramid output of layer n-1 to ensure no excessive information loss before and after sampling. This portion and subsequent subnetworks differ from the initial subnetwork in that they are UNet-like networks with a three-layer structure. Then, an upsampling operation will be performed on the processed image in the second layer and added to the Laplacian pyramid output in the n-2 layer. The operation steps of each sub-network will be repeated until the Laplacian pyramid image of the first layer is input into the *n-th* sub-network of the LMPEC algorithm. The *n-th* sub-network outputs the final corrected nameplate image.

The LMPEC algorithm consists of multiple subnetworks. The number of sub-networks is determined by the number of pyramid layers Laplace decomposed, and each sub-network weight is variable. According to the effect of global color correction and detail enhancement on the final structure, the model allocates distinct weights to each network. The sub-network of the LMPEC algorithm has a model structure similar to that of U-Net, with subsampling encoding and upsampling decoding components. It combines superficial and deep informa-tion through skip connection, providing the decoding process with more semantic informa-tion. Fig 2 illustrates the sub-networks model structure.

The LMPEC loss function consists of three components: reconstruction loss, pyramid loss, and confrontation loss. The total loss function is minimized through end-to-end model train-ing. Following is the formula for the entire loss function:

$$L = L_{rec} + L_{pyr} + L_{adv} \tag{1}$$

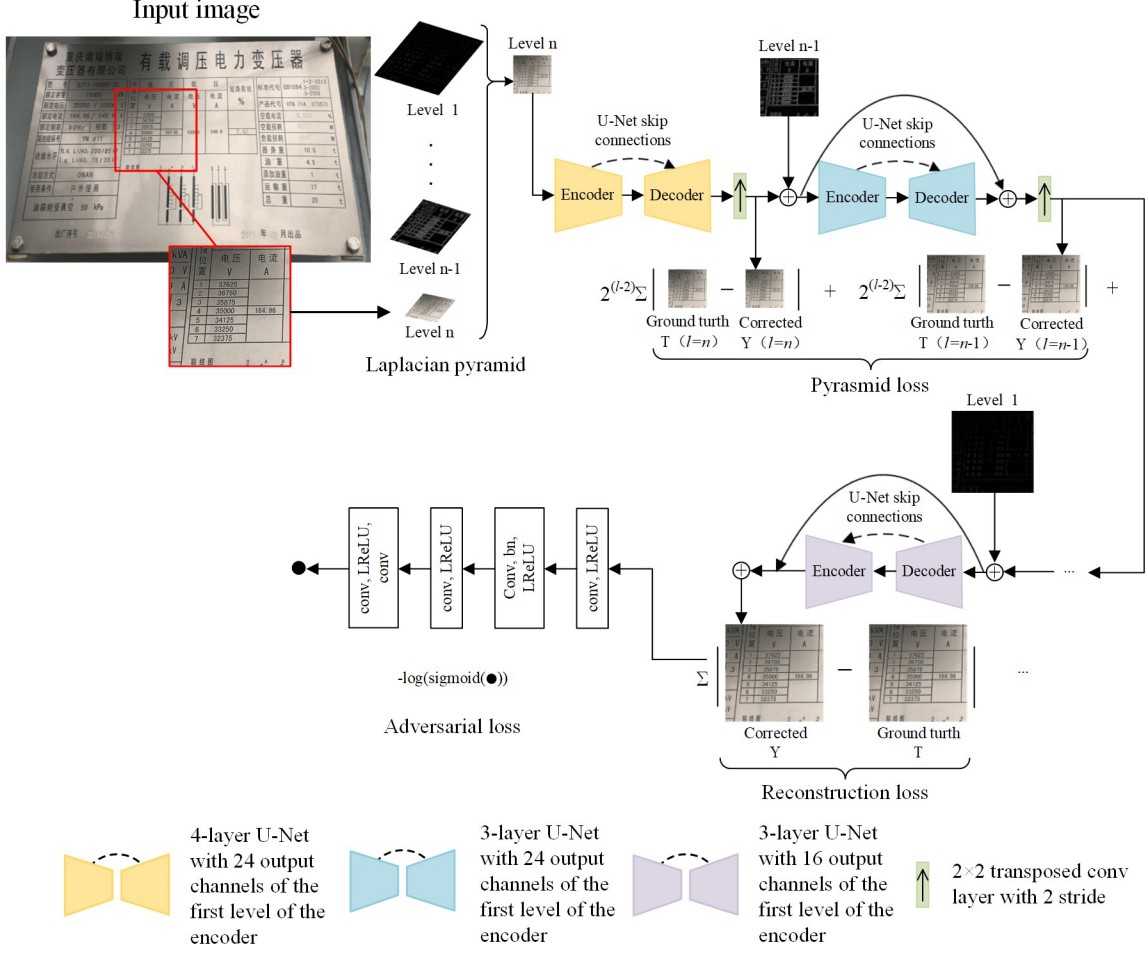

**Fig 1. Structure of the LMPEC algorithm model.**

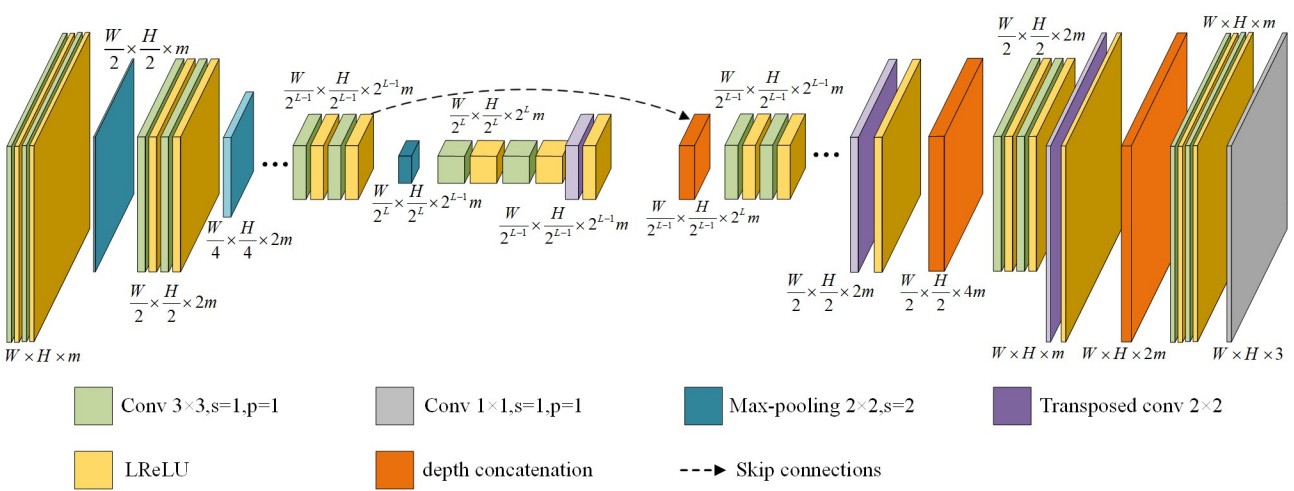

**Fig 2. Sub-network model structure in the LMPEC algorithm.**

Reconstruction loss is the L1 loss function between the standard exposure reference and model-corrected images. The reconstruction loss LREC formula is as follows:

$$L_{rec} = \sum_{p=1}^{3hw} |Y(p) - T(p)| \qquad (2)$$

Where: $h$, $w$ represents the height and width of the training image; $p$ represents the index of each pixel in the corrected image $Y$ and the standard exposure reference image $T$.

Pyramid loss is a unique loss introduced at each layer of the pyramid to direct each sub-network in the LMPEC algorithm to follow the Laplace pyramid reconstruction procedure. Pyramid loss provides a principled explanation of the tasks performed by each subnetwork and produces fewer visual artifacts than training based on reconstruction loss alone. The pyramid loss formula is as follows:

$$L_{pyr} = \sum_{l=2}^{n} 2^{(l-2)} \sum_{p=1}^{3h_1 w_1} |Y_{(l)}(p) - T_{(l)}(p)| \qquad (3)$$

Where: $T_{(l)}$ refers to the $l$-th layer of the Gaussian pyramid after double upsampling of the standard exposure image $T$; $h_1$, $w_1$ are two times the height and width of the first layer of the Laplacian pyramid of the model training image; $p$ represents the index of each pixel of $l$-th layer $Y_{(l)}$ of the model corrected image and $l$-th layer $Y_{(l)}$ of the standard exposure reference image.

Combat loss is utilized primarily to improve the perception of the corrected and reconstructed image in terms of realism and aesthetics. Similarly, combat loss is also regarded as the regularizer. In the training process of the entire LMPEC algorithm, the resistance loss is not used at the initial stage of training to accelerate the convergence speed of the network; however, once the network converges, the initial result of the network can be improved by adding the resistance loss to fine-tune the network [13]. The losses relative to the percentage of losses are as follows:

$$L_{adv} = -3hwn \log(S(D(Y))) \qquad (4)$$

Where: $S$ refers to the sigmoid function; $D$ is the discriminator DNN trained with the master network.

## LMPEC algorithm optimization

### PS-UNet++ network

**UNet++ network.** Based on the UNet network, Zhou et al. proposed the enhanced algorithm UNet++ network [14]. Compared to the network model of UNet, the redesigned network employs many skip connections. The convolution block of the skip path can reduce the semantic loss of the characteristic graph of the coder and decoder sub-network, as well as the semantic gap between the usual charts of the coder and decoder sub-network, thereby significantly enhancing the network's performance. The UNet++ network architecture is depicted in Fig 3.

The UNet++ network model consists of the encoder, the decoder, and skip connections. Like the UNet network, the encoder component is primarily used for feature extraction and subsampling of the feature map. The decoder component combines the data and generates a new feature map. In UNet++, the hopping connection is realized primarily through dense convolution blocks between the encoder and decoder. The convolution blocks can perform feature fusion of adjacent similar fault information level by level, avoiding the negative effect of direct

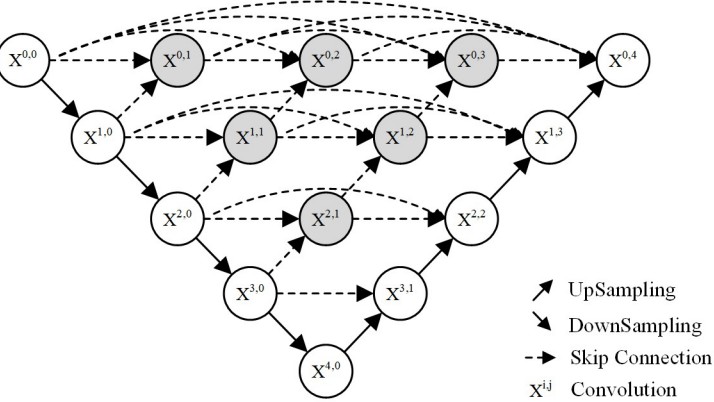

**Fig 3. Network structure diagram of UNet++.**

fusion of fault information with a significant difference in UNet so that the model can learn more accurate and detailed feature information, resulting in improved feature fusion [15].

**PS-UNet++ network based on UNet++ and efficient sub-pixel convolution.** Although the UNet++ algorithm performs marginally better than the UNet algorithm, some image information will inevitably be lost due to its frequent up-and-down sampling. Shi et al. proposed an efficient sub-pixel convolutional neural network (ECPCN) in 2016 as a deep learning upsampling method [16]. Since the technique of converting low-resolution images to high-resolution images by interpolation will result in the loss of image information and increase the model's complexity, ESPCN generates a feature image of the same size as the input by convolution on the low-resolution image and then obtains the high-resolution image by periodic screening. Consequently, sub-pixel convolution upsampling can get increasingly intricate details than interpolation upsampling. Fig 4 illustrates the subpixel convolution principle.

This paper builds a subpixel convolution upsampling module and integrates it into the UNet++ network. The merged network is now known as PS-UNet++. Fig 5 depicts the PS-U-Net++ network's architecture.

PS-UNet++ network replaces the interpolation-based upsampling module found in the original UNet++ network with a sub-pixel convolution module. Fig 6 depicts the structure of the sub-pixel convolution module. In the sub-pixel convolution upsampling module developed in this paper, the low-resolution image output from the layer *i* UNet++ network is first combined with the high-resolution feature image output from the layer *i-1* UNet++ network via

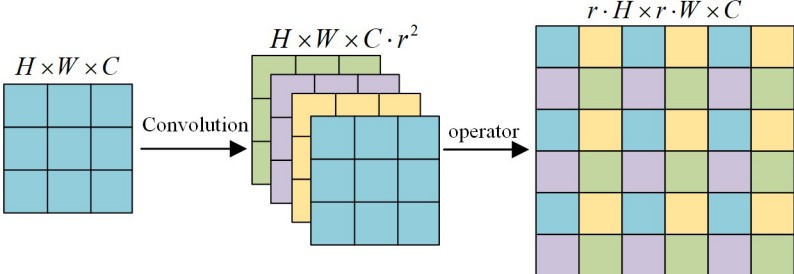

**Fig 4. Schematic diagram of subpixel convolution.**

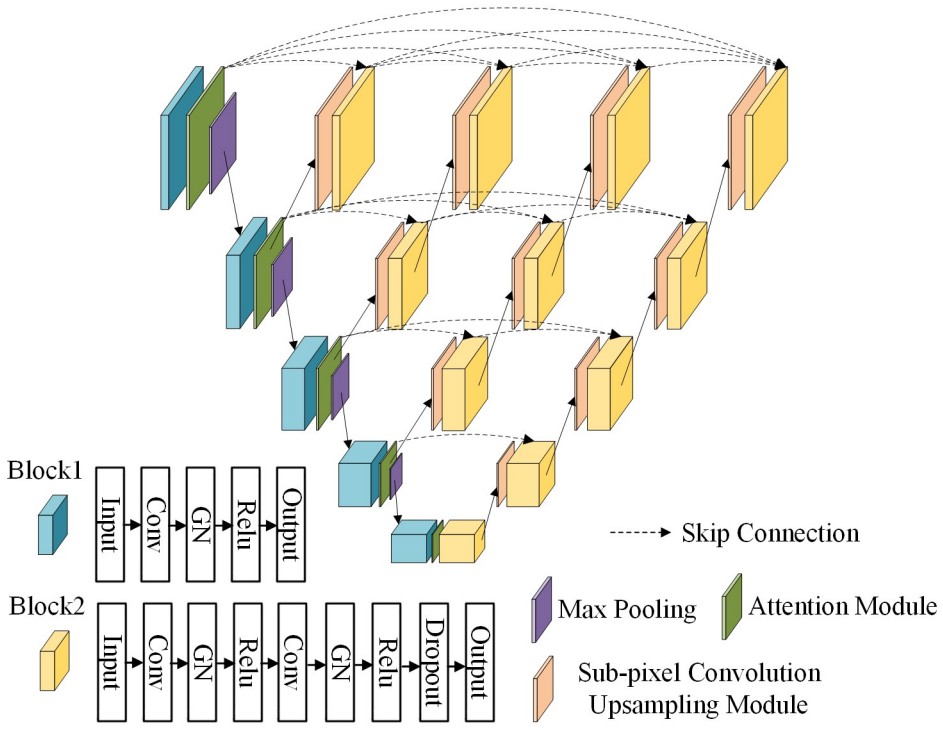

**Fig 5. PS-UNet++ model structure.**

sub-pixel convolution upsampling. Then, the number of channels of the feature image is re-adjusted using two convolution modules. Finally, the upsampled image is output.

Because there are numerous characters in the nameplate image of electrical equipment and the image quality after exposure correction directly affects the subsequent detection and

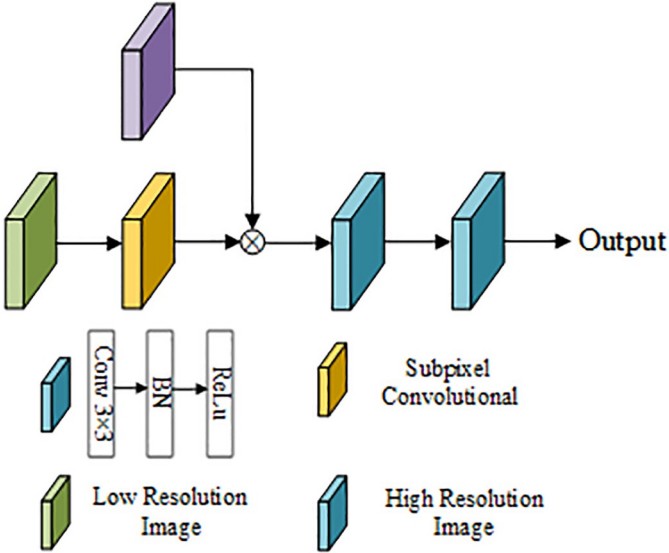

**Fig 6. Sub-pixel convolution upsampling module.**

recognition of the nameplate text of electrical equipment, it is crucial to extract the image feature information during the nameplate exposure correction process. The LMPEC algorithm relies heavily on UNet-like subnetworks for feature extraction. Compared to the UNet network, the PS UNet++ network proposed in this paper increases the complexity and computation of the model. However, it reduces the semantic difference of the model in the encoder and decoder, enabling the model to acquire richer feature information. Consequently, using a PS-UNet++ network to replace a UNet-like network as the sub-network of the LMPEC algorithm can allow the model to learn more feature information and detail information and enhance the image quality after nameplate correction.

## Loss function optimization

The reconstruction loss and pyramid loss of the LMPEC algorithm's loss function both employ L1 loss. L1 loss is also called mean absolute error (MAE), which typically refers to the average fundamental difference between the prediction model and the actual value [17]. In the LMPEC algorithm, the L1 loss of the network is the real difference between the prediction model output of each sub-network and the actual model. Following is the mathematical formula for L1 loss:

$$MAE = \frac{\sum_{i=1}^{n} |f(x_i - y_i)|}{n} \tag{5}$$

Where $f(x_i)$ and $y_i$ represent the predicted value of the $i$-th sample and its corresponding real value.

Since the derivative of the L1 loss function is constant, indicating that its gradient is stable, there will be no gradient explosion problem, and the sanction for outliers will not be punished. However, because the zero point of the L1 loss function is not differentiable and the derivative of the L1 loss function is constant, a large gradient will be obtained when the loss value is small, resulting in oscillation of the model and a poor exposure correction effect on the final electrical equipment nameplate.

Smooth L1 loss is optimized for the flaw in which L1 loss is close to zero [18]. It addresses the issue that the L1 loss function is not differentiable at the zero point when expressed as a piecewise function. The following is the formula for the Smooth L1 loss function:

$$loss(x, y) = \begin{cases} \frac{1}{n} \sum_i 0.5(x_i - y_i)^2, & if |x_i - y_i| < 1 \\ \frac{1}{n} \sum_i (|x_i - y_i| - 0.5), & otherwise \end{cases} \tag{6}$$

In general, Smooth L1 loss imposes the following two restrictions on L1 loss: first, when the absolute difference between the predicted value and the actual value is significant, the gradient value of the loss function will not be tremendous; second, when the fundamental difference between the expected value and the actual value is relatively small, the gradient value of the loss function is small enough. Consequently, using Smooth L1 loss instead of L1 loss can ensure that the difference between the predicted result and the actual result of each sub-network reconstruction loss and pyramid loss in the LMPEC algorithm is more stable, thereby facilitating the correction of the model for the exposed image of electrical equipment nameplate.

## Experimental verification

### Experimental environment

To verify the performance of this algorithm, the optimized algorithm is tested on the deep learning workstation, and the test data results are recorded and analyzed. This experiment's operating system is Windows 10 Professional, the development language is Python 3.7, the integrated development environment is Pycharm, and the deep learning framework is the GPU version of Pytorch. The CPU is an AMD EPYC 7302 with 16 cores and 32 threads and a primary frequency of 3GHz; the GPU is three NVIDIA ampere A100, and each GPU has 80G of video memory; and the system memory is 512G.

### Data set and evaluation index of nameplate exposure of electrical equipment

To evaluate the correction effect of the optimization algorithm on overexposed and underexposed images of electrical equipment nameplates, the exposure data set for electrical equipment nameplates is established in this paper. The data set comprises overexposed and underexposed images of nameplates for electrical equipment with varying exposure levels. First, 1000 original electrical equipment nameplate images are obtained through power plant photography. According to the processing method of the data set in the original LMPEC paper, professional software is used to adjust the image's exposure. For each image, use Adobe Camera Raw to simulate images with different exposure levels; the Adobe Camera Raw accurately emulates the nonlinear image rendering procedures using metadata embedded in each DNG raw file. It can affect exposure errors by setting different exposure parameters to render each image. Expressly, the exposure parameters of the original image are set to -1.5, -1, 0, +1, and +1.5 in succession to simulate the images under various exposure levels. Where -1.5, -1 indicates an underexposed image, +1.5, +1 indicates an overexposed image, and 0 indicates a standard exposure image. The final exposure data set for electrical equipment nameplates contains 5,000 images, of which 4,000 are used for training, 200 for verification, and 800 for testing. Fig 7 depicts the display of the dataset's image.

This paper selects peak signal-to-noise ratio (PSNR), structural similarity (SSIM), and image perceptual index (PI) as quantitative evaluation indicators for evaluating the corrected image to compare its effect and quality more intuitively.

PSNR is the most popular and extensively used objective image evaluation index, representing the ratio of destructive noise power to accuracy in decibels [19]. It is widely believed that the greater the PSNR value, the less the image distortion and the greater the image reconstruction effect. Following is the mathematical calculation formula for PSNR:

$$PSNR = 10 * \lg \frac{Max_I^2}{MSE} \tag{7}$$

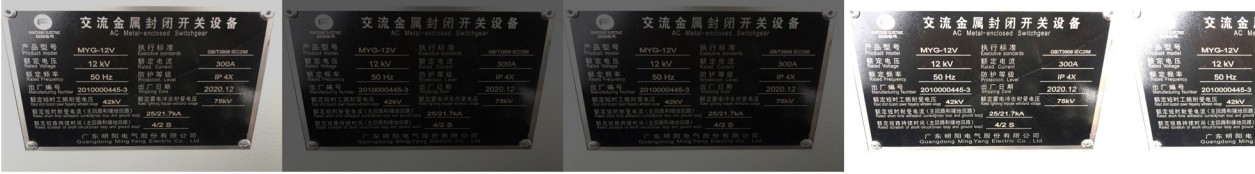

**Fig 7. Data set of nameplate exposure of electrical equipment.**

$Max_I$ represents the original data's maximum pixel value; $MSE$ represents the mean square error between the original data and the reconstructed data.

The SSIM evaluation index is a perception model that combines the structure, contrast, and luminance of two images to measure their degree of distortion and similarity [19]. SSIM can more accurately reflect the subjective feelings of individuals. The fidelity of the reconstructed image improves with increasing SSIM values. Following is the SSIM calculation formula:

$$SSIM(x, y) = \frac{(2\mu_x\mu_y + c_1)(\sigma_{xy} + c_2)}{(\mu_x^2 + \mu_y^2 + c_1)(\sigma_x^2 + \sigma_y^2 + c_2)} \tag{8}$$

Where $\mu_x$ and $\mu_y$ represent the average values of the original image $x$ and the reconstructed image $y$ respectively; $\sigma_x$ and $\sigma_y$ represent the standard deviation of $x$ and $y$ respectively; $\sigma_{xy}$ is the covariance between $x$ and $y$; $c_1$ and $c_2$ are constants.

In the 2018 PIPM-SR competition, PI was proposed and used. The PI value reflects the subjective perception of the image's quality. The lower the PI value, the higher the image fidelity and the better the subjective visual experience of the image [20]. The image quality perception index Combining the image evaluation index proposed by Ma et al. and the natural image quality evaluator (NIQE) [21, 22], the following formula is used to calculate the PI:

$$PI = \frac{1}{2}((10 - Ma + NIQE)) \tag{9}$$

## Experimental process and result analysis

**Training strategy.** Reference [23] proved that the network trained with a dataset with more considerable noise in the same noise model can denoise the noisy image. Meanwhile, the dataset with minor noise in the same model helps the network achieve better denoising performance. Based on this, Zhang et al. proposed the iterative data refinement training method to enable the model to accomplish a more effective training effect in less time. Aiming at the problem of nameplate image exposure, the training set images of the nameplate exposure data set of electrical equipment are randomly cropped one by one according to the sizes 128*128, 256*256, and 512*512. The cropped image is divided into three data sets: 128*128, 256*256, and 512*512. Lastly, the LMPEC algorithm is trained using the three data sets with the division size. In the training procedure, the Adam optimizer is utilized. The learning rate is set to 0.001, the attenuation rate to 1000, and the pitch size to 32. After training, the model's efficacy is evaluated using the test-selected image. Table 1 and Fig 8 display the test results; the input images in Fig 8 are images with exposure parameters 0, -1.5, -1, +1, and +1.5 from top to bottom.

It can be seen from the figure and table that both of the models trained by the dataset with a smaller or larger image size can correct the exposed image. Still, from the perspective of the evaluation index, the model trained by the dataset with a larger image size has a better correction effect on the exposed image.

**Table 1. Training results of data sets with different sizes.**

| Dataset Size | PSNR | SSIM | PI |
|---|---|---|---|
| 128*128 | 16.569 | 0.714 | 2.784 |
| 256*256 | 18.935 | 0.769 | 2.458 |
| 512*512 | 20.237 | 0.799 | 2.329 |

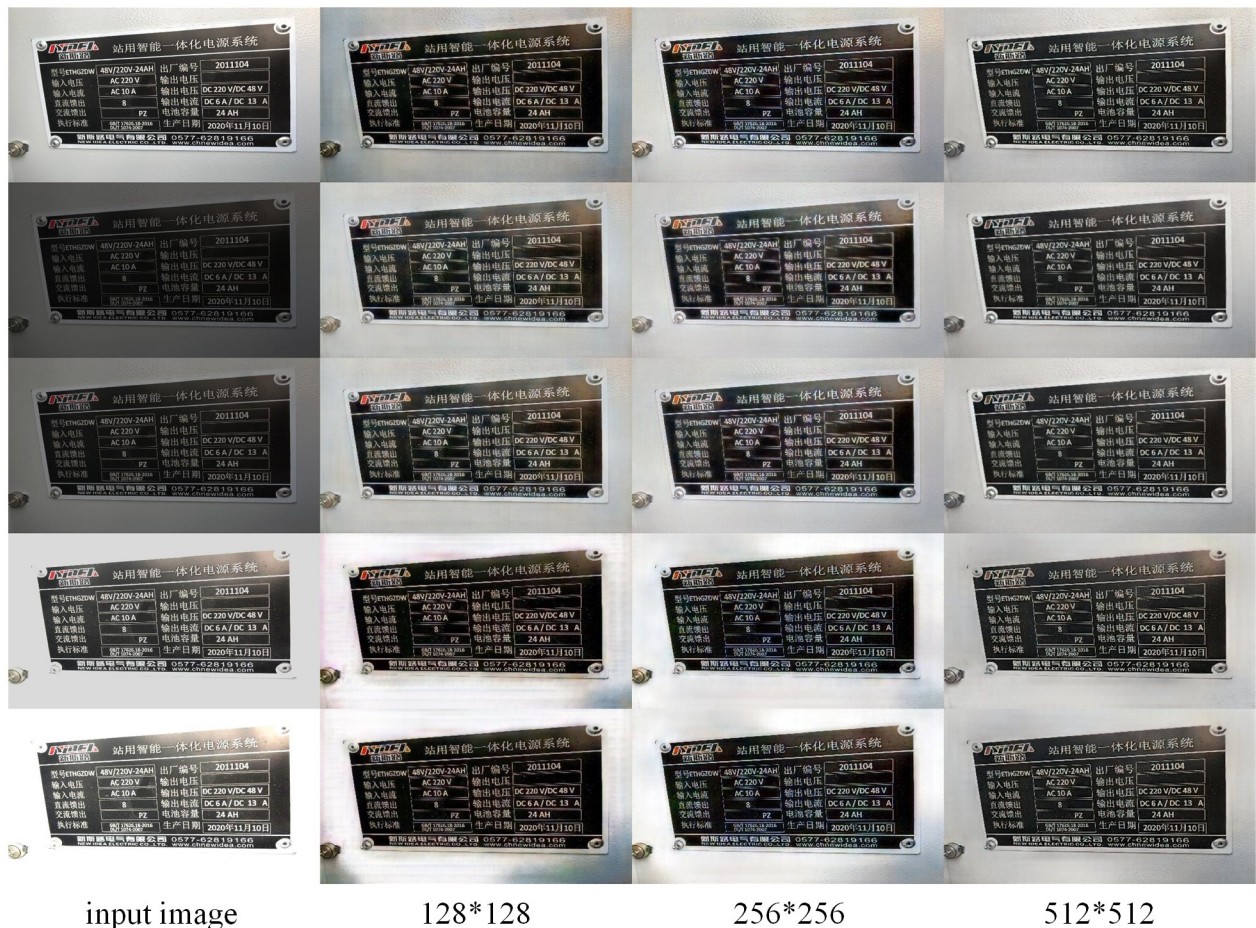

**Fig 8. Comparison of training results of data sets with different sizes.**

input image 128*128 256*256 512*512

In the area of target detection, multiscale training is commonplace. The multiscale training concept was first introduced in the reference [24]. During model training, each epoch randomly determines a size for training on a fixed number of scales. In general, the results of using small-scale images are faster but less accurate. In contrast, the test speed of using large-scale images is slower but more accurate, and the robustness of the model can be enhanced to some extent through the multi-scale training method.

Based on the multi-scale training method and the test results on datasets of varying sizes, this paper proposes a method for training the LMPEC algorithm model using datasets ranging in size from small to large. First, the nameplate image of the training set is randomly cropped to 128*128, 256*256, and 512*512, and then the cropped image is re-divided into three data sets based on its size. In the subsequent model training, three training sets are used to train the model from small to large, with slightly differing parameter settings for each size data set. Finally, the model is trained in three different scale data sets.

To compare the differences between the training method in this paper and the multi-scale training method, the original paper training method, the multi-scale training method, and the training method in this paper of the LMPEC algorithm are each tested on the electrical equipment nameplate exposure data set. Table 2 contains the test results. After using the multi-scale

**Table 2. Test results of different training methods.**

| Training Method | PSNR | SSIM | PI |
|---|---|---|---|
| Original Training Method | 21.082 | 0.812 | 2.220 |
| Multiscale Training Method | 21.236 | 0.803 | 2.237 |
| Our Training Method | 21.526 | 0.827 | 2.203 |

training method, the PSNR value increased, but the SSIM value and PI value decreased. After using the training method in this paper, PSNR, SSIM, and PI are improved to a certain extent, demonstrating that the training method in this paper can improve the robustness and anti-interference of the LMPEC model, as well as the correction effect of the model on the exposure nameplate.

**Ablation experiment.** To observe the influence of each optimization module in the LMPEC optimization algorithm, ablation experiments were conducted from three perspectives: using PS-UNet++ to replace the sub-network, using the Smooth L1 loss function, and implementing new training methods. The experiment utilized the data set of nameplate exposure of electrical equipment, and the results are shown in Table 3.

It can be seen from the data in the table that the evaluation indices have been enhanced to some degree after successively adding various optimization methods to the LMPEC algorithm. After using PS-UNet++ to replace the subnet, the PSNR and SSIM values increased by 2.59% and 3.08%, respectively, and the PI value decreased by 5.19%. After using the Smooth L1 loss function, PSNR and SSIM increased by 1.38% and 1.08%, respectively, and PI decreased by 1.96%. After using the new training strategy, the PSNR and SSIM values increased by 1.55% and 0.83%, respectively, and the PI value decreased by 1.02%. According to the test results, it can be demonstrated that the three optimization methods used in this paper for the LMPEC algorithm are effective in the data set of electrical equipment nameplate exposure, thereby improving the model's exposure correction effect.

**Comparative experiment.** Adam optimizer is utilized in the training procedure of the optimization algorithm presented in this paper. The batch size is 32, the initial learning rate is 0.001, and the decay rate is set to 1000. Throughout the training of 128*128 datasets, the initial learning rate is maintained. During the training of 256*256 and 512*512 datasets, the learning rate is halved compared to the initial learning rate, and it is halved again every ten cycles of training until the loss curve of the model tends to stabilize.

To verify the effect of the algorithm, several exposure correction algorithms are selected and compared with the experimental results. In addition to the histogram equalization (HE), local adaptive histogram equalization (CLAHE), Retinexnet, HQEC algorithm, and zero-DCE algorithm mentioned in the introduction, Enlightengan [25] (deep light enhancement without paired supervision) algorithm and Mbllen [26] (low light image/video enhancement using CNN) algorithm are selected as the comparison algorithm of the optimization algorithm in

**Table 3. Ablation test results of the data set of electrical equipment nameplate exposure.**

| LMPEC | PS-UNet++ | Smooth L1 | Our Training Method | PSNR | SSIM | PI |
|---|---|---|---|---|---|---|
| √ | | | | 21.082 | 0.812 | 2.210 |
| √ | √ | | | 21.629 | 0.837 | 2.096 |
| √ | √ | √ | | 21.927 | 0.846 | 2.055 |
| √ | √ | √ | √ | 22.266 | 0.853 | 2.034 |

**Table 4. Training and test results of different algorithms on uneven exposure data set of electrical equipment nameplate.**

| Method | PSNR | SSIM | PI |
|---|---|---|---|
| HE4 | 17.223 | 0.702 | 2.670 |
| CLAHE5 | 16.721 | 0.698 | 2.885 |
| RetinexNet 7 | 15.881 | 0.623 | 2.912 |
| HQEC8 | 19.004 | 0.765 | 2.315 |
| Zero-DCE9 | 14.426 | 0.630 | 2.869 |
| EnlightenGAN 25 | 19.547 | 0.781 | 2.294 |
| MBLLEN26 | 16.527 | 0.704 | 2.527 |
| LMPEC11 | 21.082 | 0.812 | 2.210 |
| Ours | 22.266 | 0.853 | 2.034 |

this paper. Enlightengan is a network for enhancing images based on the generation counter-measure network Gan. This network can effectively utilize unpaired images for training. The Mbllen network is primarily used to improve images with limited illumination. The network employs multiple subnetworks to enhance the extracted feature map and combines the multi-branch output results to produce the final enhanced image. In the comparative experiment, the networks above were trained and evaluated using the data set of electrical equipment nameplate exposure, and the results are presented in Table 4.

On the data set of electrical equipment nameplate exposure, the PSNR, SSIM, and PI of the optimization algorithm presented in this paper are 22.266, 0.853, and 2.034, respectively, the best performance among all comparison algorithms. In addition, compared to the LMPEC algorithm, the optimization algorithm in this paper has increased PSNR and SSIM by 5.6% and 5.1%, respectively, and decreased the PI value by 7.96%, indicating that the optimization algorithm in this paper has a performance improvement over the original algorithm, and has a better effect on the over exposure and under exposure image correction of electrical equipment nameplate.

Fig 9 depicts the correction comparison of this paper's optimized algorithm and other algorithms on the exposure data set of electrical equipment nameplate, where (a), (b), (c), and (d) correspond to the effect comparison after image correction of exposure parameters -1.5, -1, +1, and +1.5, respectively.

The comparison chart reveals that although different algorithms can correct overexposed and underexposed images of electrical equipment nameplates, there are differences in the correction effect. Most algorithms can demonstrate a decent correction effect for underexposed images, whereas the correction effect for overexposed images is subpar. In this paper, the optimization algorithm of LMPEC shows effective correction of overexposed and underexposed electrical equipment nameplate images.

Because the nameplate contains a large quantity of character information, the image quality of the corrected nameplate image is crucial for subsequent text detection and recognition. Therefore, This paper selects the overexposed image with an exposure parameter of +1.5 in Fig 10, enlarges the local text information on the corrected nameplate image, and compares the algorithm again. Fig 10 illustrates the comparative outcomes. The comparison chart demonstrates that, among the various algorithms evaluated, the method presented in this paper has the highest image quality after correcting the nameplate exposure image, and the text on the image is visible, which provides great convenience for the subsequent text detection and recognition of the nameplate of electrical equipment.

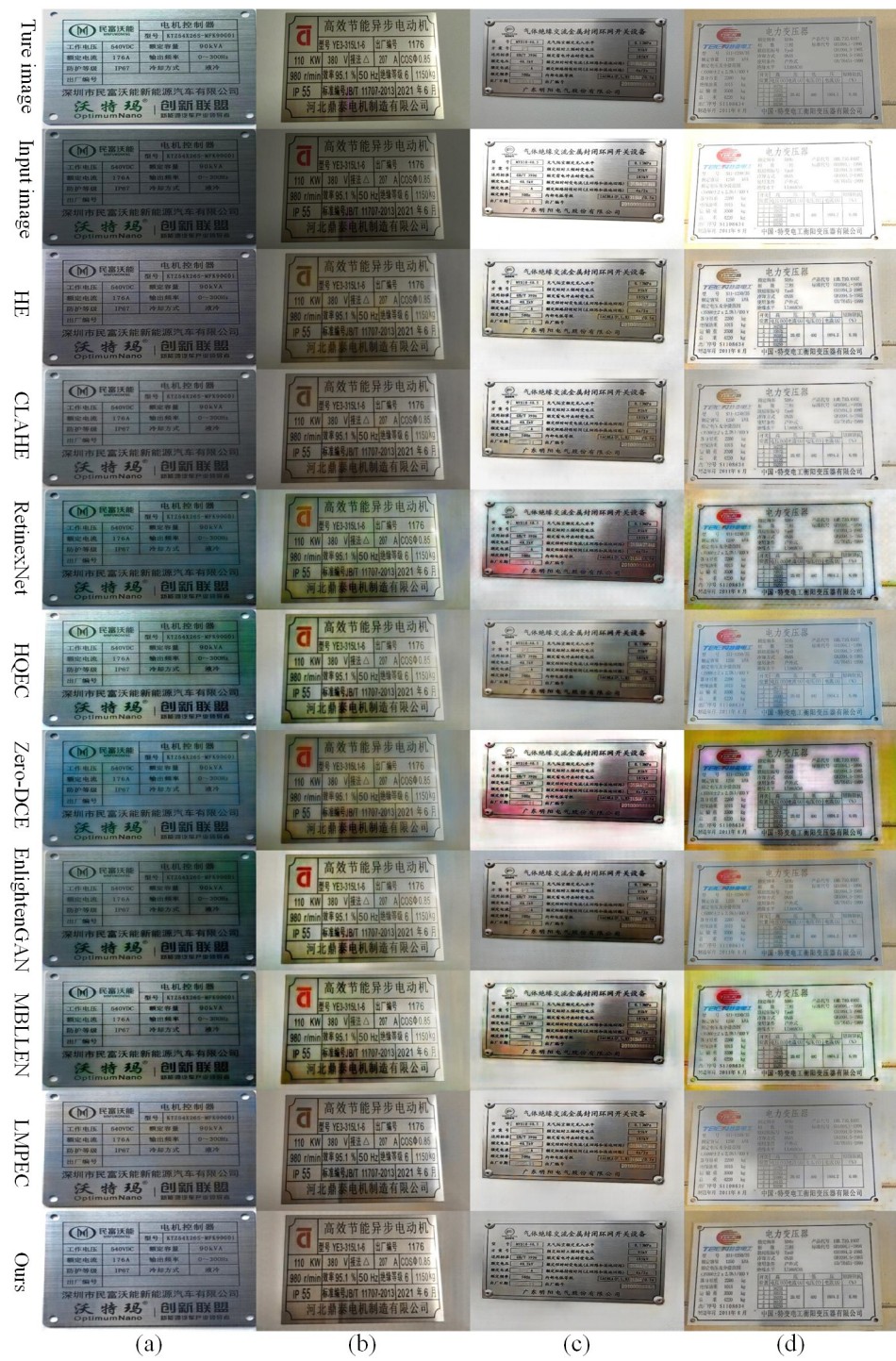

**Fig 9. Comparison of correction results of different algorithms on the data set of electrical equipment nameplate exposure.**

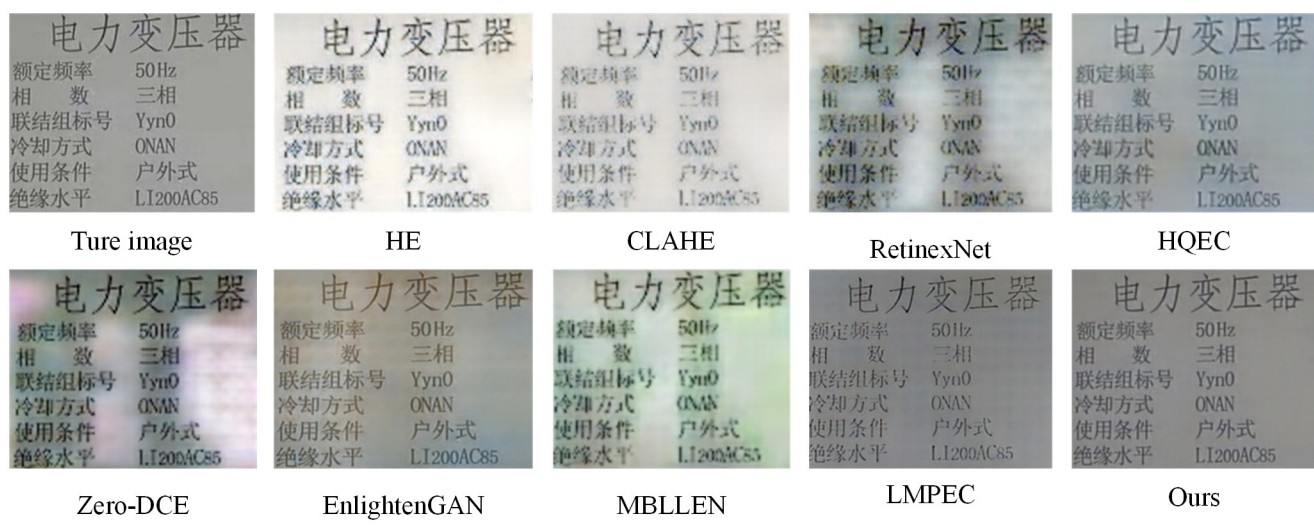

**Fig 10. Comparison of details of different algorithms.**

To further compare the performance of the enhanced algorithm presented in this paper, the natural scene exposure data set proposed in the original LMPEC paper is used to test the algorithm again. 24,330 images are included in the data set, with 17,675 used for training, 750 for verification, and 5,905 for testing. The obtained results are shown in Table 5. The evaluation index data in the table indicates that the optimization algorithm in this paper continues to perform the best on the natural scene exposure data set. Compared to the original LMPEC algorithm, the optimization algorithm in this paper has an improvement on the evaluation indices PSNR, SSIM, and PI, which further demonstrates that the optimization of the LMPEC algorithm in this paper is effective.

## Summary

This paper proposes an optimization algorithm based on the LMPEC algorithm to address the overexposure and underexposure problems of the nameplate image of electrical equipment. This optimization algorithm builds a PS UNet++ network based on the sub-pixel convolution method and UNet++ network. Then, it uses this network to replace all UNet-like subnetworks in LMPEC so that the model can learn more details in the electrical equipment nameplate and

**Table 5. Training test results of different algorithms on natural scene exposure dataset.**

| method | PSNR | SSIM | PI |
|---|---|---|---|
| HE4 | 16.204 | 0.672 | 2.423 |
| CLAHE5 | 15.784 | 0.641 | 2.514 |
| RetinexNet7 | 14.325 | 0.630 | 3.357 |
| HQEC8 | 16.839 | 0.692 | 2.623 |
| Zero-DCE9 | 13.168 | 0.569 | 3.052 |
| EnlightenGAN25 | 17.359 | 0.702 | 2.341 |
| MBLLEN26 | 15.929 | 0.683 | 2.436 |
| LMPEC11 | 19.402 | 0.732 | 2.255 |
| Ours | 20.257 | 0.775 | 2.126 |

improve the quality of the final correction image of over-exposure and under-exposure nameplate. In addition, the Smooth L1 loss function is substituted for the L1 loss function to prevent model oscillation and influence the model's effect on the exposed image. During model training, the enhanced multi-scale training method is used to strengthen the model's robustness further. Compared with other algorithms, the optimized algorithm has a more significant correction effect on the exposed image, and the corrected image has uniform brightness, less noise, and better image quality. In addition, the objective evaluation indices PSNR, SSIM, and PI demonstrate that the algorithm presented in this paper is superior, and the efficacy of the optimization algorithm presented in this paper is enhanced compared to the original LMPEC algorithm. However, because the improved method of the model presented in this paper increases the model's complexity and calculation, and the data set of the nameplate of electrical equipment is insufficient, future work will concentrate on optimizing the model's complexity and expanding the data set so that the model can demonstrate a better exposure correction effect.

## Supporting information

**S1 Dataset.**
(ZIP)

## Author Contributions

**Conceptualization:** Yanxi Liu.

**Data curation:** Yanxi Liu.

**Funding acquisition:** Hao Wu.

**Investigation:** Yanxi Liu.

**Methodology:** Yanxi Liu, Zhongyang Jin.

**Resources:** Hao Wu, Yanxi Liu.

**Software:** Yanxi Liu.

**Supervision:** Hao Wu, Yanxi Liu.

**Validation:** Hao Wu, Yanxi Liu, Zhongyang Jin.

**Writing – original draft:** Yanxi Liu.

**Writing – review & editing:** Hao Wu, Zhongyang Jin, Yuan Zhou.

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
