## [Decision Letter · Decision Letter 0]

2 Jan 2024

PONE-D-23-32825Exposure Image Correction of Electrical Equipment Nameplate Based on LMPEC Optimization AlgorithmPLOS ONE

Dear Dr. yanxi,

Thank you for submitting your manuscript to PLOS ONE. After careful consideration, we feel that it has merit but does not fully meet PLOS ONE’s publication criteria as it currently stands. Therefore, we invite you to submit a revised version of the manuscript that addresses the points raised during the review process.

We look forward to receiving your revised manuscript.

Kind regards,

Richard Jiang

Academic Editor

PLOS ONE

“This study received partial funding from the Sichuan Provincial Department of Science and Technology projects 2021YFG0313 and 2022YFS0518.”

6. Please upload a copy of Figures 1, 2, 3, 4, 5, 6, 7, 8, 9 and 10 to which you refer in your text. If the figure is no longer to be included as part of the submission please remove all reference to it within the text.

7. We note that Supplementary figures 1, 7, 8, 9 and 10 in your submission contain copyrighted images. All PLOS content is published under the Creative Commons Attribution License (CC BY 4.0), which means that the manuscript, images, and Supporting Information files will be freely available online, and any third party is permitted to access, download, copy, distribute, and use these materials in any way, even commercially, with proper attribution. For more information, see our copyright guidelines: http://journals.plos.org/plosone/s/licenses-and-copyright.

1. You may seek permission from the original copyright holder of Supplementary figures 1, 7, 8, 9 and 10 to publish the content specifically under the CC BY 4.0 license.

Reviewers' comments:

Reviewer's Responses to Questions

**Comments to the Author**

1. Is the manuscript technically sound, and do the data support the conclusions?

Reviewer #1: Yes

Reviewer #2: Yes

Reviewer #3: Partly

2. Has the statistical analysis been performed appropriately and rigorously? 

Reviewer #1: Yes

Reviewer #2: Yes

Reviewer #3: No

3. Have the authors made all data underlying the findings in their manuscript fully available?

Reviewer #1: Yes

Reviewer #2: Yes

Reviewer #3: Yes

4. Is the manuscript presented in an intelligible fashion and written in standard English?

Reviewer #1: Yes

Reviewer #2: Yes

Reviewer #3: Yes

5. Review Comments to the Author

Reviewer #1: 1. General Comments:

In this paper, the LMPEC algorithm is proposed to optimize the overexposure and underexposure problem of electrical equipment nameplate images. In constructing the PS-UNet++ network, all three and four layers of Unet-like subnetworks are replaced. The training process and loss function are also optimized and enhanced. And by testing the constructed electrical equipment nameplate exposure dataset, the results show that the algorithm in this paper can perform better exposure correction for both overexposed and underexposed images of electrical equipment nameplates. However, there are several issues with this manuscript. It should therefore be systematically revised. The following comments may help enhance the quality of this manuscript.

2. List the inadequacies of current researches and the innovations of this article point by point in the introduction. Here, the reviewer recommends the following articles for the author's references:

Nameplate text detection of power plant electrical equipment based on improved EAST algorithm

Hybrid semantic segmentation for tunnel lining cracks based on swin transformer and convolutional neural network

3. Detailed Comments:

1. Starting with the second paragraph of each section, the first sentence should be indented by 2 characters.

2. In section 3.3, please explain the reason for selecting the 7 m width of the pit slope as the subject of the study, i.e., how the value of the slope width was determined.

3. In Figure 8, the exposure parameters for each row of images should be indicated.

4. The image before correction should be added in Figure 9 and used to compare with the effect image after correction in order to analyze the correction effect in both underexposure and overexposure cases.

5. The content of Table 5 shows that the optimization algorithm proposed in this paper improves the evaluation indices, but the evaluation indices of this algorithm in the table do not differ much from the improved values of other algorithms, while the complexity and computational volume of this algorithm increases relatively, is it possible to assess whether this algorithmic improvement is meaningful in relation to the increase in complexity?

Reviewer #2: An optimization algorithm based on the LMPEC algorithm is proposed to rectify the nameplate image. The list of comments is suggested to be concerned to improve the paper:

1. The writing should be improved to add the readability.

2. The motivation of this paper should be summarized in the introduction clearly.

3. The contributions of this paper should be summarized in the introduction clearly.

4. Explain the rationality of the chosen data sets.

Reviewer #3: This paper proposed an optimization algorithm based on the LMPEC algorithm to rectify the nameplate image to address the problem that overexposure and underexposure of the nameplate image of electrical equipment. The paper itself has certain engineering practical application value. However, this paper also has the following problems:

1) There are duplicates in the name of the paper.

2) In the introduction, the core problem of image recognition is not clearly explained. It is difficult to judge the theoretical contribution of this paper.

3) There is some unreason ability in the structure of the paper, such as separating the pictures and words of the paper, which brings some difficulties to the reading.

4) The data source of the images adopted in the paper is not clear enough. At the same time the paper lacks the analysis of the recognition image. The paper uses some methods, but these methods from the point of view of solving the problem, whether necessary. It is suggested that the author should make further explanation.

6. PLOS authors have the option to publish the peer review history of their article (what does this mean?). If published, this will include your full peer review and any attached files.

Reviewer #1: No

Reviewer #2: No

Reviewer #3: No

---

## [Author Response · Author response to Decision Letter 0]

27 Jan 2024

Dear Editor and Reviewers:

Thank you for your letter and for the Review experts’ comments concerning our manuscript. Those comments are all valuable and very helpful for revising and improving our paper, as well as the important guiding significance to our research. We have studied comments carefully and have made corrections which we hope meet with approval. Revised portions are highlighted in yellow on the Revised Manuscript with Track Changes. The main corrections in the paper and responses to the reviewers' comments are as follows:

Response to Academic Editor:

All the documents I have submitted are in accordance with the requirements of PLOE ONE.

I am a graduate student at Sichuan University of Science & Engineering, and my work in this paper has been approved by the Artificial Intelligence Laboratory of Sichuan University of Science & Engineering.

3. Please note that PLOS ONE has specific guidelines on code sharing for submissions in which author-generated code underpins the findings in the manuscript. In these cases, all author-generated code must be made available without restrictions upon publication of the work.

The complete code and version of the manuscript have been uploaded to the system together.

I will ensure the correct funding number when resubmitting.

5. Please state what role the funders took in the study.

6. Please upload a copy of Figures 1, 2, 3, 4, 5, 6, 7, 8, 9 and 10 to which you refer in your text. If the figure is no longer to be included as part of the submission please remove all reference to it within the text.

I have re uploaded all the images in the correct format.

7. We note that Supplementary figures 1, 7, 8, 9 and 10 in your submission contain copyrighted images.

The images mentioned by the academic editor were all taken under the permission of the relevant factory, so there is no copyright issue with the images I used in the paper.

8. Please include captions for your Supporting Information files at the end of your manuscript, and update any in-text citations to match accordingly.

All supporting information related to the article has been uploaded, and all references have been updated.

Response to expert reviewer comments：

Reviewer #1's comments：

1. Starting with the second paragraph of each section, the first sentence should be indented by 2 characters.

The format of this paper strictly follows the writing requirements of PLOS ONE, and there is no requirement from PLOS ONE to indent the first sentence of each paragraph in the original manuscript.

2. In section 3.3, please explain the reason for selecting the 7 m width of the pit slope as the subject of the study, i.e., how the value of the slope width was determined.

There is no section 3.3 and research content on mine slope here, which may be an incorrect comment.

3. In Figure 8, the exposure parameters for each row of images should be indicated.

The revised version of the manuscript has added a description of the input image exposure parameters before Figure 8.

4. The image before correction should be added in Figure 9 and used to compare with the effect image after correction in order to analyze the correction effect in both underexposure and overexposure cases.

Figure 9 contains the image before correction, which is in the second row of Figure 9.

5. The content of Table 5 shows that the optimization algorithm proposed in this paper improves the evaluation indices, but the evaluation indices of this algorithm in the table do not differ much from the improved values of other algorithms, while the complexity and computational volume of this algorithm increases relatively, is it possible to assess whether this algorithmic improvement is meaningful in relation to the increase in complexity?

Compared to the algorithm before optimization, the model in this paper has a certain improvement in complexity and computational complexity. The model in this paper is mainly composed of several UNet++ networks. Compared with most other network models, the UNet++ network has a simple structure and low computational complexity. Therefore, even though the improved model has increased computational complexity compared to the original model, overall, all the complexity and computational complexity of the model are still relatively small, and the model can still be easily trained. It is worthwhile to exchange a slight increase in complexity for an average performance improvement of about 6%.

Reviewer #2's comments：

1. The writing should be improved to add the readability.

The revised version of the manuscript has adjusted some sentences in the paper to increase readability.

2. The motivation of this paper should be summarized in the introduction clearly.

The revised version of the manuscript has added explanatory sentences on writing motivation in the introduction.

3. The contributions of this paper should be summarized in the introduction clearly.

The revised version of the manuscript has an introduction section that has added the contribution of this paper.

4. Explain the rationality of the chosen data sets.

The revised version of the manuscript has added paragraphs on the sources and processing methods of the dataset to demonstrate its rationality.

Reviewer #3's comments：

1. There are duplicates in the name of the paper.

The revised version of the manuscript has removed "optimization" from the title, and the revised title is Exposure Image Correction of Electrical Equipment Nameplate Based on LMPEC Algorithm.

2. In the introduction, the core problem of image recognition is not clearly explained. It is difficult to judge the theoretical contribution of this paper.

The revised version of the manuscript has an introduction section that has added the contribution of this paper.

3. There is some unreason ability in the structure of the paper, such as separating the pictures and words of the paper, which brings some difficulties to the reading.

The format of this paper strictly follows the writing requirements of PLOS ONE, and PLOS ONE requires that papers and images be submitted separately.

4. The data source of the images adopted in the paper is not clear enough. At the same time the paper lacks the analysis of the recognition image. The paper uses some methods, but these methods from the point of view of solving the problem, whether necessary. It is suggested that the author should make further explanation.

For the question with datasets, the revised version of the manuscript has added paragraphs on the sources and processing methods of the dataset to demonstrate its rationality. 

For the question of the necessity of this study, the main research content of this paper is the exposure correction of electrical equipment nameplates. However, in addition to this study, the author's research team has also conducted relevant research on detecting and recognizing text information on electrical equipment nameplates. We found that when there is insufficient or overexposure in the image of the electrical equipment nameplate, it can have a certain impact on detecting nameplate text. As shown in the figure below, our team uses an improved algorithm based on the EAST algorithm to perform text detection on images with exposure parameters of -1.5, 0,and +1.5, respectively. From the figure, the nameplate image under standard exposure has fewer missed and false detections compared to under exposure and over exposure images. Therefore, we conducted related research on exposure image correction of electrical equipment nameplates. In addition, the revised version of the manuscript has added relevant explanations on the necessity of this study in the introduction section.

---

## [Decision Letter · Decision Letter 1]

6 Mar 2024

Exposure Image Correction of Electrical Equipment Nameplate Based on the LMPEC Algorithm

PONE-D-23-32825R1

Dear Dr. yanxi,

We’re pleased to inform you that your manuscript has been judged scientifically suitable for publication and will be formally accepted for publication once it meets all outstanding technical requirements.

Kind regards,

Richard Jiang

Academic Editor

PLOS ONE

Additional Editor Comments (optional):

Reviewers' comments:

Reviewer's Responses to Questions

**Comments to the Author**

1. If the authors have adequately addressed your comments raised in a previous round of review and you feel that this manuscript is now acceptable for publication, you may indicate that here to bypass the “Comments to the Author” section, enter your conflict of interest statement in the “Confidential to Editor” section, and submit your "Accept" recommendation.

Reviewer #2: All comments have been addressed

Reviewer #3: All comments have been addressed

2. Is the manuscript technically sound, and do the data support the conclusions?

Reviewer #2: Yes

Reviewer #3: Yes

3. Has the statistical analysis been performed appropriately and rigorously? 

Reviewer #2: Yes

Reviewer #3: Yes

4. Have the authors made all data underlying the findings in their manuscript fully available?

Reviewer #2: Yes

Reviewer #3: Yes

5. Is the manuscript presented in an intelligible fashion and written in standard English?

Reviewer #2: Yes

Reviewer #3: Yes

6. Review Comments to the Author

Reviewer #2: The authors have answered all the concerns of the reviewer. It is ready for the publication. There is no further comment.

Reviewer #3: The author has answered all the questions raised by the reviewers.The current manuscript is suitable for publication.

7. PLOS authors have the option to publish the peer review history of their article (what does this mean?). If published, this will include your full peer review and any attached files.

Reviewer #2: No

Reviewer #3: No

---

## [Editor Report · Acceptance letter]

23 Mar 2024

PONE-D-23-32825R1 

PLOS ONE

Dear Dr. Liu, 

I'm pleased to inform you that your manuscript has been deemed suitable for publication in PLOS ONE. Congratulations! Your manuscript is now being handed over to our production team.

Kind regards, 

on behalf of

Dr. Richard Jiang 

Academic Editor

PLOS ONE